# Robust Regression with Density Power Divergence: Theory, Comparisons, and Data Analysis

**DOI:** 10.3390/e22040399

**Published:** 2020-03-31

**Authors:** Marco Riani, Anthony C. Atkinson, Aldo Corbellini, Domenico Perrotta

**Affiliations:** 1Dipartimento di Scienze Economiche e Aziendale and Interdepartmental Centre for Robust Statistics, Università di Parma, l43125 Parma, Italy; mriani@unipr.it (M.R.); aldo.corbellini@unipr.it (A.C.); 2The London School of Economics, London WC2A 2AE, UK; a.c.atkinson@lse.ac.uk; 3European Commission, Joint Research Centre, 21027 Ispra, Italy

**Keywords:** estimation of α, monitoring, numerical minimization, S-estimation, Tukey’s biweight

## Abstract

Minimum density power divergence estimation provides a general framework for robust statistics, depending on a parameter α, which determines the robustness properties of the method. The usual estimation method is numerical minimization of the power divergence. The paper considers the special case of linear regression. We developed an alternative estimation procedure using the methods of S-estimation. The rho function so obtained is proportional to one minus a suitably scaled normal density raised to the power α. We used the theory of S-estimation to determine the asymptotic efficiency and breakdown point for this new form of S-estimation. Two sets of comparisons were made. In one, S power divergence is compared with other S-estimators using four distinct rho functions. Plots of efficiency against breakdown point show that the properties of S power divergence are close to those of Tukey’s biweight. The second set of comparisons is between S power divergence estimation and numerical minimization. Monitoring these two procedures in terms of breakdown point shows that the numerical minimization yields a procedure with larger robust residuals and a lower empirical breakdown point, thus providing an estimate of α leading to more efficient parameter estimates.

## 1. Introduction

Basu et al. [1] introduced a general form of robust estimation based on minimizing a density power divergence. The family of procedures, and so the robustness properties, depend on the value of a parameter α. In this paper, we consider normal theory regression. We use standard methods for the analysis of robust procedures, in particular S-estimation (Riani et al. [2]), to find the theoretical breakdown point and efficiency of power divergence regression as a function of α. We use these results to make comparisons with theoretical properties of other robust methods, for example, S-estimation using Tukey’s biweight. We introduce a data-driven method for the estimation of α from monitoring residuals over a range of values of α and so find the empirical efficiency and breakdown point of power density estimation for several regression examples. One surprising conclusion is that, for normal theory models, the rho function for the power divergence is one minus a suitably scaled standard normal density raised to the power α.

The paper is structured as follows. The next section introduces minimum density power divergence estimation and the related estimating equations for normal theory linear regression. The important problem of estimating α is mentioned. The first part of Section 3 reviews S-estimation in the linear regression model, and the second part, Section 3.2, rewrites power divergence estimation of the regression parameter β in the form of S-estimation, derives the rho function, and so finds the asymptotic breakdown point (bdp) of the procedure. Section 3.2.2 gives the asymptotic efficiency of this S-estimation at the Gaussian model and finds the weight function used in fitting data. Comparisons are given with some well known rho and weight functions. In Section 4, plots of asymptotic efficiency against asymptotic bdp are used to compare the properties of several S-estimators, including Tukey’s biweight. Section 5 compares methods through the analysis of data. An alternative to S power divergence is the original suggestion of Basu et al. [1] to use Brute Force (BF) minimization (our acronym, not theirs). Comparisons on simulated and real data show the superiority of BF power divergence to the S-estimator. In particular, monitoring the plots of residuals as α varies may lead to a clear indication of the minimum value of α for which a robust fit is obtained. Thus, the empirical breakdown point of BF power divergence estimation can be found, leading to the most efficient robust estimation for each specific data set.

## 2. Minimum Density Power Divergence Estimation

Basu et al. [1] define the power divergence between two densities f(z) and g(z), a function of a single parameter α, as
(1)$$\begin{array}{ccc} d_{\alpha }\left\{g\left(z\right),f\left(z\right)\right\} & = & \int \left\{f^{1+\alpha }\left(z\right)-\left(1+\frac{1}{\alpha }\right)f^{\alpha }\left(z\right)g\left(z\right)+\frac{1}{\alpha }g^{1+\alpha }\left(z\right)\right\}dz,\;\alpha >0\\ d_{0}\left\{g\left(z\right),f\left(z\right)\right\} & = & \int g\left(z\right)\log\left\{\frac{g(z)}{f(z)}\right\}dz. \end{array}$$

The parameter α controls the trade-off between efficiency and robustness for the power divergence estimator. The limit as α→0 is a version of the Kullback-Leibler divergence. The value α=1 leads to squared $L_{2}$ estimation, an analysis of which is given by Scott [3].

Let *g* be the density function of the process generating the data. Given an independent and identically distributed sample $y_{1},\ldots ,y_{n}$ is available from *G*, Basu et al. [1] model the unknown g(z) with the density $f_{\theta }\left(y\right)$ by minimizing $d_{\alpha }\left\{g\left(z\right),f_{\theta }\left(y\right)\right\}$. Since the third term of the divergence is independent of θ, the power divergence estimator of θ can be found by minimizing
(2)$$\int f_{\theta }^{1+\alpha }\left(z\right)dz-\left(1+\frac{1}{\alpha }\right)\frac{1}{n}\sum_{i=1}^{n}f_{\theta }^{\alpha }\left(y_{i}\right),$$
in which the empirical distribution $G_{n}$ is used to approximate the unknown distribution *G*, thus avoiding the necessity for density estimation.

Basu et al. [1] develop their method only for random samples from the normal, exponential and Poisson distributions. For the normal distribution, Equation (Equation 2) is minimized over both the mean μ and the variance $\sigma^{2}$. The extension to normal theory regression models is in Ghosh and Basu [4].

As usual in a regression framework, we define $y_{i}$ to be the response variable, which is related to the values of a set of p-1 explanatory variables $x_{i1},\ldots ,x_{ip-1}$ by the relationship
(3)$$y_{i}=\beta^{\prime }x_{i}+\epsilon_{i}\;i=1,\ldots ,n,$$
where, including an intercept, $\beta^{\prime }=\left(\beta_{0},\beta_{1},\ldots ,\beta_{p-1}\right)$ and $x_{i}={\left(1,x_{i1},\ldots ,x_{ip-1}\right)}^{\prime }$. Let $\sigma^{2}=\mathrm{var}\left(\epsilon_{i}\right)$, which is assumed to be constant for all i=1,…,n. We also take the quantities in $x_{i}$ to be fixed and assume that $x_{1},\ldots ,x_{n}$ are not collinear. The case p=1 corresponds to that of a univariate response without predictors. We call σ the scale of the distribution of the error term $\epsilon_{i}$, when its density takes the form
$$\sigma^{-1}f\left(\frac{\epsilon }{\sigma }\right).$$

When *f* is the normal distribution with mean, as in Equation (Equation 3), and variance $\sigma^{2}$, Durio and Isaia [5] and Ghosh and Basu [4] show that the function, as in Equation (Equation 2), to be minimized becomes
(4)$$\frac{1}{{\left(2\pi \right)}^{\alpha /2}\sigma^{\alpha }\sqrt{1+\alpha }}-\frac{1+\alpha }{\alpha }\frac{1}{{\left(2\pi \right)}^{\alpha /2}\sigma^{\alpha }}\frac{1}{n}\sum_{i=1}^{n}e^{-\alpha {\left(y_{i}-x_{i}^{\prime }\beta \right)}^{2}/2\sigma^{2}}.$$

The partial derivative of Equation (Equation 4), with respect to $\beta_{j}$, provides the estimating equation for β:(5)$$\sum_{i=1}^{n}x_{ij}\left(y_{i}-x_{i}^{\prime }\beta \right)e^{-\alpha {\left(y_{i}-x_{i}^{\prime }\beta \right)}^{2}/2\sigma^{2}},\;\left(j=1,\ldots ,p\right).$$

When α=0, Equation (Equation 5) becomes the equation for non-robust ordinary least squares. For α>0 we have weighted least squares of the kind associated in the next section with M estimation. Ghosh and Basu [4] also give the estimating equation for $\sigma^{2}$ which we will however not be using in our theoretical development.

An important aspect is the estimation of α. Durio and Isaia [5] test for changes in the estimates of the parameters β as a function of α, while Warwick and Jones [6] and Ghosh and Basu [7] estimate the mean squared error of the parameter estimates as α changes. In Section 5, we monitor changes in the pattern of residuals to choose the minimum value of α for which a robust fit is obtained, so leading to the most efficient parameter estimates.

## 3. Robust Regression

### 3.1. M and S Estimation

Basu et al. [1] find estimates of the parameters of the linear model by simultaneous minimization of Equation (Equation 4) as a function of β and $\sigma^{2}$. In this section, we recall the theory of M and S estimation, which we use in Section 3.2 to describe properties of the S power divergence estimator. In Section 5, we provide a numerical comparison of the BF minimization and S-estimation approaches.

The M-estimator of the regression parameters, which is scale equivariant (i.e., independent of the units of measurement), is defined by
(6)$${\hat{\beta }}_{M}=\min_{\beta \in {ℜ}^{p}}\sum_{i=1}^{n}\rho \left(\frac{r_{i}}{s}\right),$$
where $r_{i}=y_{i}-\beta^{\prime }x_{i}$ is the *i*-th residual and ρ is a function with suitable properties and *s* is an estimate of σ. For least squares $\rho \left(x\right)=x^{2}.$ For robust estimation $\rho \left(x\right)<x^{2}$ for sufficiently large absolute values of *x*. We also write $r_{i}\left(\beta \right)$ to emphasize the dependence of $r_{i}$ on β.

These definitions do not depend on how σ is estimated. Clearly, if we want to keep the M-estimate robust, *s* should also be a robust estimate. We assume that the same ρ is used in the estimation of β and σ, which is customary in practice. In order to have a consistent scale estimate for normally distributed observations, we require
(7)$$E_{\Phi_{0,1}}\left[\rho \left(\frac{r_{i}}{s}\right)\right]=K,$$
where $\Phi_{0,1}$ is the cdf of the standard normal distribution. To see consistency, notice that $E_{\Phi_{0,1}}\left(\rho \right)=K$ implies
$$\frac{E_{\Phi_{0,\sigma^{2}}}\left[\rho \right]}{K}=\frac{K\sigma^{2}}{K}=\sigma^{2}.$$

An M-estimator of scale in Equation (Equation 3), say *s*, is defined to be the solution to the equation
(8)$$\frac{1}{n}\sum_{i=1}^{n}\rho \left(\frac{r_{i}}{s}\right)=\frac{1}{n}\sum_{i=1}^{n}\rho \left(\frac{y_{i}-\beta^{\prime }x_{i}}{s}\right)=K.$$

Equation (Equation 8) is solved, at least in principle, among all $\left(\beta ,\sigma \right)\in {ℜ}^{p}\times \left(0,\infty \right)$, where 0<K<supρ. Rousseeuw and Yohai [8] defined S-estimators by minimization of the dispersion *s* of the residuals
(9)$${\hat{\beta }}_{S}=\min_{\beta \in {ℜ}^{p}}s\left\{r_{1}\left(\beta \right),\ldots ,r_{n}\left(\beta \right)\right\}$$
with final scale estimate
$${\hat{\sigma }}_{S}=s\left\{r_{1}\left({\hat{\beta }}_{S}\right),\ldots ,r_{n}\left({\hat{\beta }}_{S}\right)\right\}.$$

The dispersion *s* is defined as the solution of Equation (Equation 8). The S-estimates, therefore, can be thought as self-scaled M-estimates whose scale is estimated simultaneously with the regression parameters. Note, in fact, that when the scale and the regression estimates are simultaneously estimated, S-estimators for regression also satisfy (for example, Maronna et al. [9], p. 131)
(10)$${\hat{\beta }}_{S}=\min_{\beta \in {ℜ}^{p}}\sum_{i=1}^{n}\rho \left(\frac{r_{i}}{s}\right).$$

The estimator of β in Equation (Equation 9) is called an S-estimator because it is derived from a scale statistic in an implicit way.

The function ρ is the key to many important properties of M and S estimates. Rousseeuw and Leroy [10] (p. 139) show that, if the function ρ satisfies the following conditions:It is symmetric and continuously differentiable, and ρ(0)=0;there exists a c>0 such that ρ is strictly increasing on [0,c] and constant on [c,∞); andit is such that
(11)K/ρ(c)=bdpwith0<bdp≤0.5,

then the asymptotic breakdown point of the *S*-estimator tends to bdp when n→∞. Note that if ρ(c) is normalized in such a way that supρ(c)=1, the constant *K* becomes exactly equal to the breakdown point of the *S*-estimator.

### 3.2. S Estimation for Power Divergence Regression

#### 3.2.1. The Breakdown Point and the Rho Function

The function ρ is used in the estimation of β for a given estimate *s*. With x=r/s it follows from the function to be minimized in Equation (Equation 4) that $\rho \left(x\right)\propto -\exp\left(-\alpha x^{2}/2\right)$. If we scale this function so that $\sup\rho_{\alpha }\left(x\right)=1$ and $\rho_{\alpha }\left(0\right)=0$, we obtain
(12)$$\rho_{\alpha }\left(x\right)=1-\exp\left(-\alpha x^{2}/2\right).$$

This is a trivial reparameterization of an otherwise unreferenced rho function attributed to Welsh.

The panels of Figure 1 show plots of $\rho_{\alpha }\left(x\right)$ for several values of α. For α=1, the efficiency is 0.65, and the breakdown point is 0.29. As α decreases, the procedure becomes less robust but more efficient. Table 1 gives values of α, bdp, and *eff* for three frequently used values of each quantity; these values being given in bold. The left-hand panel of Figure 1 is for the three bold values of bdp, and the right-hand panel for the three values of *eff*. The rho functions for high efficiency are appreciably flatter than those for high bdp.

Since $\rho_{\alpha }$ is scaled, the breakdown point, bdp, is given by $E_{\Phi_{0,1}}\left[\rho_{\alpha }\left(x\right)\right]$. Then,
(13)$$\begin{array}{ccc} E_{\Phi_{0,1}}\left[\rho_{\alpha }\left(x\right)\right] & = & 1-E\left[\exp(-\alpha x^{2}/2)\right],\\ \; & = & 1-\int \exp(-\alpha x^{2}/2)dx,\\ \; & = & 1-{\left(2\pi \right)}^{\alpha /2}\int \phi_{0,1}^{\alpha }\left(x\right)\phi_{0,1}\left(x\right)dx. \end{array}$$

From the useful general expression in Section 3.2 of Basu et al. [11] that
$$\int \phi_{m,s}^{\alpha }\left(x\right)\phi_{c,d}\left(x\right)dx=\frac{\exp\left[-\alpha {\left(c-m\right)}^{2}/\left\{2\left(s^{2}+\alpha d^{2}\right)\right\}\right]}{{\left(2\pi \right)}^{\alpha /2}s^{\alpha }{\left(1+\frac{\alpha d^{2}}{s^{2}}\right)}^{0.5}},$$
we obtain
(14)$$E_{\Phi_{0,1}}=\mathrm{bdp}=1-\frac{1}{\sqrt{1+\alpha }}.$$

Our expression for the breakdown point comes from S-estimation, reflecting breakdown in the estimate of β under the customary assumption that σ is known. This is different from the value of
(15)$$\frac{\alpha }{{\left(1+\alpha \right)}^{3/2}}$$
in Section 3.2 of Basu et al. [11], who consider the joint breakdown of the estimates of β and σ when “location explodes” and “scale implodes”. While the expression in Equation (Equation 14) increases monotonically in the interval α=[0,3], Equation (Equation 15) increases monotonically in the smaller interval α=[0,2] and then slightly decreases.

To fit a model to data, we specify the desired asymptotic breakdown point, when the value of α from inverting the expression in Equation (Equation 14) is
$$\alpha =\frac{1}{{\left(1-\mathrm{bdp}\right)}^{2}}-1.$$

For example, for 50% breakdown, α=3.

#### 3.2.2. Efficiency, the Psi Function and the Influence Function

Other basic properties of the robust estimator follow from derivatives of $\rho_{\alpha }\left(x\right)$. For power density
$$\psi_{\alpha }\left(x\right)=\rho_{\alpha }^{\prime }\left(x\right)=\alpha x\exp\left(-\alpha x^{2}/2\right)$$
and
$$\psi_{\alpha }^{\prime }\left(x\right)=\alpha \left(1-\alpha x^{2}\right)\exp\left(-\alpha x^{2}/2\right).$$

Figure 2 shows, for three values of α, a plot of $\psi_{\alpha }\left(x\right)$ (which is proportional to the Influence Function, see Maronna et al. [9] (p. 123)). As α decreases, the figure shows the curve becomes flatter.

From, for example, Rousseeuw and Leroy [10] (p. 142), the asymptotic efficiency *eff* of the S-estimator at the Gaussian model is
(16)$$\mathrm{eff}=\frac{{\left\{\int \psi^{\prime }\left(x\right)d\Phi \left(x\right)\right\}}^{2}}{\int \psi^{2}\left(x\right)d\Phi \left(x\right)}.$$

For $\rho_{\alpha }\left(x\right)$,
(17)$$E\left[\psi_{\alpha }^{2}\left(x\right)\right]=\alpha^{2}{\left(2\pi \right)}^{\alpha /2}\int x^{2}\phi_{0,1}^{2\alpha x+1}dx.$$

Since
$$\int x^{2}\phi_{0,1}^{n}dx=\frac{1}{n^{3}{\left(2\pi \right)}^{n-1}},$$

Equation (Equation 17) becomes
$$E\left[\psi_{\alpha }^{2}\left(x\right)\right]=\alpha^{2}\frac{1}{{\left(2\alpha +1\right)}^{3}}.$$

To find the numerator of the efficiency
(18)$$\begin{array}{ccc} E[\psi_{\alpha }^{\prime }\left(x\right)] & = & \alpha {\left(2\pi \right)}^{\alpha /2}\int \phi_{0,1}^{\alpha +1}dx-\alpha^{2}{\left(2\pi \right)}^{\alpha /2}\int x^{2}\phi_{0,1}^{\alpha +1}dx,\\ \; & = & \frac{\alpha }{\sqrt{1+\alpha }}-\frac{\alpha^{2}}{\sqrt{{\left(1+\alpha \right)}^{3}}},\\ \; & = & \frac{\alpha }{\sqrt{{\left(1+\alpha \right)}^{3}}}. \end{array}$$

Combining these pieces, we obtain
(19)$$\mathrm{eff}=\frac{\sqrt{(1+2\alpha }{)}^{3}}{{\left(1+\alpha \right)}^{3}},$$
agreeing with the expression for the asymptotic variance of the estimate of the mean μ of a univariate normal sample given in Section 4.2 of Basu et al. [1], a few values of which are tabulated in their Table 1. Inversion of Equation (Equation 19) yields
$$\alpha =(1-F+\sqrt{1-F})/F,$$
where $F={\mathrm{eff}}^{2/3}$.

The algorithm for S-estimation is complicated, involving weighted regression. Rousseeuw and Leroy [10] (pp. 207–208) provide a sketch. More details are in Salibian-Barrera and Yohai [12]. A central part is weighted regression, with weights
w(x)=ψ(x)/x.

Figure 3 plots the weight functions for power divergence and five other rho functions: Tukey’s biweight [13], Hampel’s [14] (p. 150), Huber’s [15], the optimal (Yohai and Zamar [16]), and hyperbolic tangent (Hampel et al. [14] (p. 328)), all scaled to have efficiency 0.95.

Details of the functions are in the Appendix A. The similarity of the power divergence weights to those of the Tukey biweight is outstanding, although the biweight is exactly zero at x=c, which in this case is equal to 4.6851. For this *x* coordinate, the power divergence weight (when *eff* = 0.95) is 0.0851. Both have a curved shape for small values of |x|, unlike the Hampel and hyperbolic weights. We note that the procedure for finding the tuning constant α for the power divergence estimator, given a prefixed value of breakdown point or efficiency, is not iterative. This is distinct from all the other rho functions listed above (apart from that of Huber), for which iterative procedures are required.

## 4. Comparisons of Asymptotic Properties

The basic properties of S power divergence are the asymptotic breakdown point, as in Equation (Equation 14), and the asymptotic efficiency, as in Equation (Equation 19). Figure 4 shows these two properties as functions of α over the range 0≤α≤3. As bdp increases from zero towards 0.5, *eff* decreases from 1 to 0.2894. These are generic shapes for robust estimators, quantifying the trade-off between robustness and efficiency. Figure 5 shows plots of efficiency against breakdown point for S power divergence and four of the other ρ functions of Figure 3 (the Huber function being excluded because it has a zero breakdown point). In order to generate these curves, we fix a particular value of breakdown point and find the associated tuning constant α for PD or *c* for the other estimators (the details are in the Appendix). In the case of the Hampel ρ functions, the three extra parameters $c_{1}$, $c_{2}$, and $c_{3}$ have been set equal to 2, 4, and 8. For the hyperbolic tangent estimator the extra parameter *k*, which reflects the log of the change of variance sensitivity of the *M*-estimator, has been set equal to 4.5. Given the value of the tuning constant, we found the corresponding value of the efficiency.

It is clear from the figure that the general asymptotic performance of the five methods is similar. The optimal function is best for small bdp but worst for values slightly larger than 0.25. The situation for Hampel is the reverse, being worst for small bdp and best for bdp values above approximately 0.4. For small bdp, the power divergence is the second worst but behaves much like the hyperbolic and biweight functions for larger values of bdp. For 50% bdp (as the inset in the figure shows), the ordering is (we give the exact numbers in parenthesis) hyperbolic (0.3019), Hampel (0.2924), power divergence (0.2894), biweight (0.2868), and last the optimal (0.2428). Hössjer [17] proves that, for normal theory linear models, the maximum efficiency when bdp = 0.5 is 0.329.

Some further insight into the balance between breakdown point and efficiency comes from varying the parameters of the Hampel and hyperbolic functions. In Figure 5, the parameters for the Hampel were $c_{1}=2$, $c_{2}=4$, and $c_{3}=8$. The left-hand panel of Figure 6 compares the breakdown point and efficiency of Hampel’s rho function with these values to those when $c_{1}=1.5$, $c_{2}=3.5$, and $c_{3}=8$. The original procedure is better for breakdown point less than around 0.3, with the modified version being slightly better for larger values. For the hyperbolic rho function in the right-hand panel the freely variable parameter, other than *c*, is *k*. The curves for three values of *k* are shown in the right-hand panel of Figure 6. The difference is largest for small values of bdp, when k=6 has the highest efficiency. In other words, imposing a looser constraint in the change of variance parameter produces higher efficiency for small values of bdp. For breakdown points near 0.5, the order is reversed, with k=6 being the least efficient, although, in this region, the differences are less than for low bdp. The conclusion from this figure reinforces that from Figure 5; no one rho function has the highest breakdown point and efficiency over the whole range of bdp from 0 to 0.5. These results also implicitly show that the choice of the ρ function is not a crucial aspect since all (provided they are bounded) have similar behavior in terms of breakdown point and efficiency. These theoretical results are in line with the empirical findings in Salini et al. [18], where it is shown that the size of the test for outlier detection is much more affected by the choice of the requested level of efficiency or breakdown point than by the choice of the ρ function.

It is hard to reconcile the conclusions from these graphs with the statement in the opening paragraph of Jones et al. [19] that “quite small values of α were found to afford considerable robustness while retaining very high efficiency relative to maximum likelihood”. Although it may be argued that S power divergence has good properties as a robust procedure, the figure shows that these fully agree with those for other S estimators. We now turn from asymptotics to data analysis to allow non-asymptotic comparisons and analysis of the ‘brute force’ approach to power divergence estimation.

## 5. Monitoring and Comparisons with Data

In order to compare the finite sample properties of robust estimators in regression, Riani et al. [20] introduced the idea of monitoring the properties of robust analyses as tuning constants are changed. For power divergence, this would be the value of α, or equivalently changes in nominal values of bdp or *eff*, which are how the range of monitored values was specified for other ρ functions. The most incisive information comes from looking at displays of residuals. Typically, for contaminated data, these display many outliers for very robust analyses, which suddenly are much reduced in magnitude at a specific value of the tuning constant. At this point, the procedure becomes close to maximum likelihood including the outliers. The sharp transition between the two regions allows estimation of the empirical breakdown point and so to the robust analysis with the highest efficiency. The monitoring process starts with bdp=0.5, which is the maximum fraction of contamination that an affine equivariant estimator can resist.

To illustrate this structure, we re-analyze regression data from Atkinson and Riani [21] (Table A2) comparing S power divergence with the BF version, using numerical minimization. We start monitoring from a bdp of 50% and use the very robust version of Least Median of Squares regression (Rousseeuw [22]) to provide initial estimates of β and $\sigma^{2}$. After this initial minimization for α=3, successive minimizations for lower values of α start from the estimates for the immediately higher value of α.

The regression data consist of 60 response observations and three explanatory variables. The scatter-plot matrix of the data does not reveal any outlying observations. The upper panel of Figure 7 is the monitoring plot of the residuals for BF power divergence as α goes from 3 to 0. There is a very clear transition from the robust analysis in the left-hand part of the plot to the non-robust analysis in the right-hand part, which occurs just before bdp = 0.21, giving an empirical breakdown point of 0.23. What is striking about this figure, apart from the clear transition point, is the distinct near constancy of the residuals in the two parts of the plot.

The lower panel of Figure 7 is the same plot but for the analysis using S power divergence. The conclusion is similar, with an empirical breakdown point of 0.27, higher than that in the upper panel; BF therefore provides more efficient estimates. Although the residuals in the non-robust right-hand part are constant, those from the robust analysis decrease in magnitude as the analysis becomes less robust. This effect is caused by the gradual increase in the estimate of $\sigma^{2}$ as the analysis becomes less robust. A monitoring plot of the two estimates of σ is in the left-hand panel of Figure 8. The BF estimate is indeed virtually constant up to a bdp of nearly 0.3, increasing more rapidly to bdp = 0.2 with a jump corresponding to the switch from robust to non-robust analysis. At this point, it is close to that from S-estimation, which has been continually increasing. Both estimates of course coincide when bdp = 0, that is, for non-robust least squares.

These plots show the importance of the empirical breakdown point, found as α, and hence bdp, decrease. We monitor at values $\alpha_{i},i=1,\ldots ,n_{\alpha }$, corresponding to breakdown values bdp${}_{i}$. In our examples, $n_{\alpha }=50.$ At each *i*, we calculate a property of the fit, $P_{i}$ and find the difference $D_{i}=|P_{i}-P_{i-1}$|. Let the empirical breakdown point be bdp${}^{*}$. Then,

**Definition** **1.**
*The empirical breakdown point bdp${}^{*}$ = bdp${}_{i^{*}}$, where*
$$i^{*}=\arg\max D_{i},i=1,\ldots ,n_{\alpha }-1.$$


Some choices of the property $P_{i}$ are

The residual sum of squares.Changes in the parameter estimates ${\hat{\beta }}_{i}$ or $\hat{\sigma }$.Measures of correlation between successive sets of residuals, rather than the sum of squares (Riani et al. [20]).

This definition is for fixed finite *n*. If there are *m* outliers with responses $y_{j}^{\prime }=y_{j}+\Delta_{j},j=1,\ldots ,m$, determination of bdp${}^{*}$ is sharp as $\Delta_{j}\to \infty .$ As $\Delta_{j}\to 0,$ a threshold should be applied in the calculation of $i^{*}$.

We ran a number of simulations and studied the monitoring plots. For a data set of 100 observations without outliers, the trajectories of the residuals were smooth and uneventful, although a similar structure was observed to that of Figure 7: the residuals from BF were sensibly constant until around α=1 and then began gently to become less extreme. On the other hand, the S residuals steadily decreased in magnitude. The plot of the estimates of σ was similar to that of the left-hand panel of Figure 8. As is correct in the absence of outliers, neither plot of residuals nor σ indicated the need for robust analysis.

When the outliers in our simulations were very remote, both methods clearly indicated the outliers, although the monitoring plot for S estimation, unlike that using BF, did not show a sharp transition between two regions. The challenge for robust methods is when the outliers are less remote. As an example, we again simulated 100 observations with $\sigma^{2}=1$, but now a value of 5 was added to 20 responses. The two panels of Figure 9 show the resulting monitoring plots. Both display the same set of scaled residuals for 50% bdp, although those from BF are larger in magnitude. BF shows relatively sharp transitions at a breakdown point of 0.16, whereas S estimation shows a gradual decrease in the magnitude of the residuals as bdp (α) decreases. The right-hand panel of Figure 8 plots the two estimates of σ. As in the results for the regression data, the estimate from S-estimation increases gradually as bdp decreases, but the BF estimates are sensibly constant until a bdp around 0.16, when there is a distinct increase due to non-robust estimation.

Our results in Section 3.2.1 and Section 4 indicate the close relationship between Tukey’s biweight and the power density rho functions. This is illustrated by the plot for S estimation using the biweight on these data, which we do not show here, which is indistinguishable from that using the power divergence ρ.

As a final larger data example, we analyze 509 observations on the amount spent by loyalty card holders at a supermarket chain in Northern Italy, introduced by Atkinson and Riani [23], who recommended a Box-Cox transformation for the response with λ=1/3. Perrotta et al. [24] showed that a value of λ=0.4 is to be preferred. We used this value in our analysis. The monitoring plot of residuals from BF power divergence is in Figure 10. It shows stable trajectories of the residuals for many values of α. A change starts around bdp = 0.17, indicating this as the empirical bdp. Again, S power divergence, which we do not show, reveals the same extreme observations, but fails to provide a sharp transition, so that the empirical breakdown point for efficient analysis is again not easily determined.

## 6. Discussion

We have used the estimating equation for the linear parameters β to recast power divergence estimation in the context of S-estimation. This leads straightforwardly to calculations of asymptotic bdp and efficiency. This form of the power density estimate has asymptotic properties close to those of S estimation using Tukey’s biweight.

An alternative to power divergence S-estimation is brute-force numerical minimization. The non-asymptotic comparison of the two procedures has been performed with monitoring plots of residuals as bdp varies, providing fits changing from very robust to maximum likelihood. S power divergence estimation has properties very similar to those of S-estimation with Tukey’s biweight. In both, there is often a smooth decrease in the magnitude of the residuals as bdp decreases. On the other hand, BF minimization produces monitoring plots which show a clearer break between robust and non-robust fits, leading to estimation of an empirical breakdown point and so to the most efficient robust estimates.

One conclusion is that BF estimation provides more informative analyses than power density S-estimation. However, the results of monitoring regression in Riani et al. [20] show that the comparative behavior of estimators depends on the particular data set being analyzed. Figure 7 shows that S-estimation may produce monitoring plots with a sharp change, and further examples are in Riani et al. [20]. Other methods providing a sharp change, and so guidance to efficient analysis, are the Forward Search [25] and Least Trimmed Squares [22]. It remains to be seen how BF power divergence compares with these other methods, both statistically and on larger, more complicated models, such as linear mixed models, generalized linear models, or nonlinear models.

## Figures and Tables

**Figure 1 entropy-22-00399-f001:**
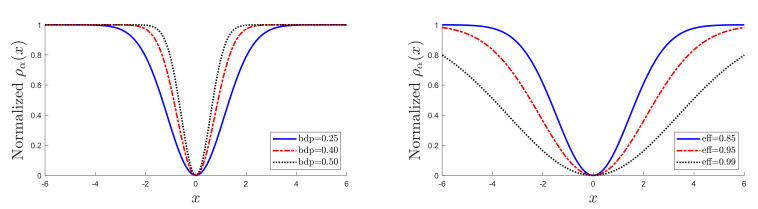
Dependence of $\rho_{\alpha }\left(x\right)$ on α, for frequently used values of robustness properties in Table 1. Left-hand panel, three values of breakdown point (bdp); right-hand panel, three values of *eff*.

**Figure 2 entropy-22-00399-f002:**
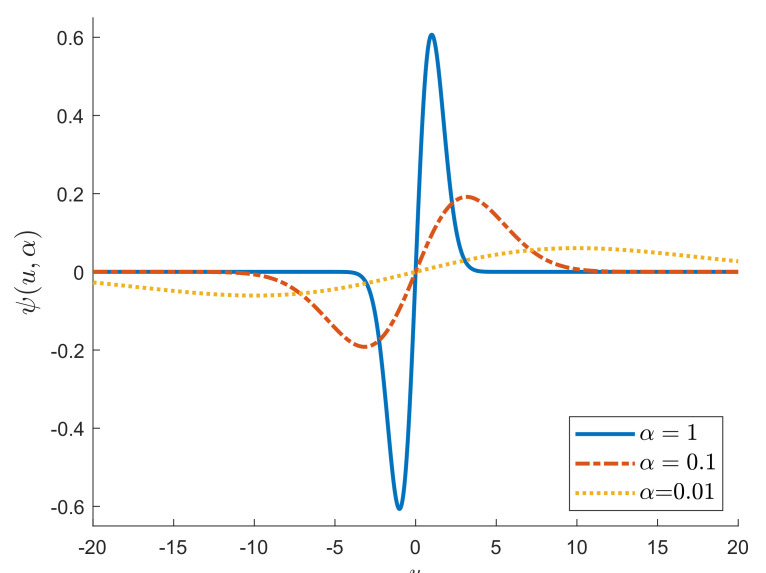
S power divergence; ψ function, proportional to the influence function.

**Figure 3 entropy-22-00399-f003:**
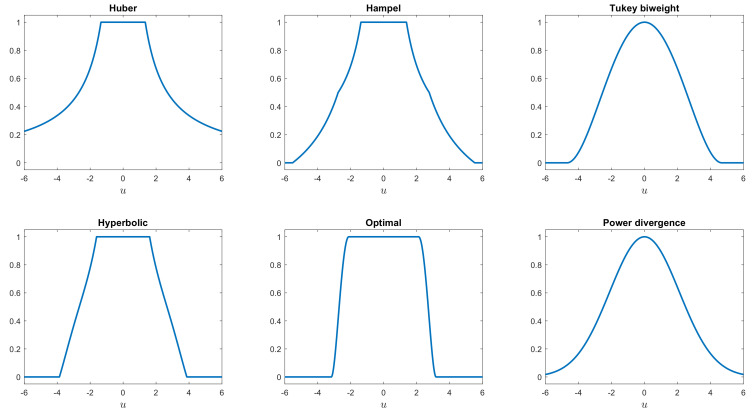
The weight function ψ(x)/x for six S-estimators.

**Figure 4 entropy-22-00399-f004:**
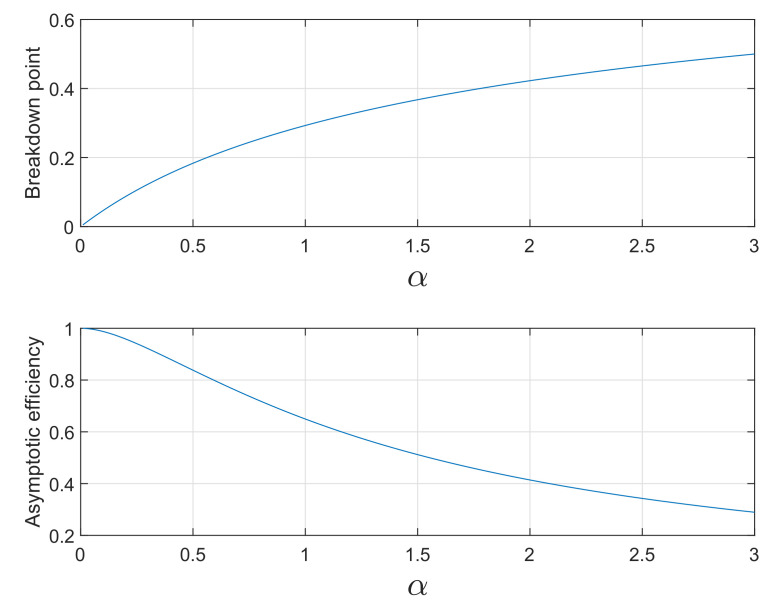
S power divergence: breakdown point and efficiency as functions of α.

**Figure 5 entropy-22-00399-f005:**
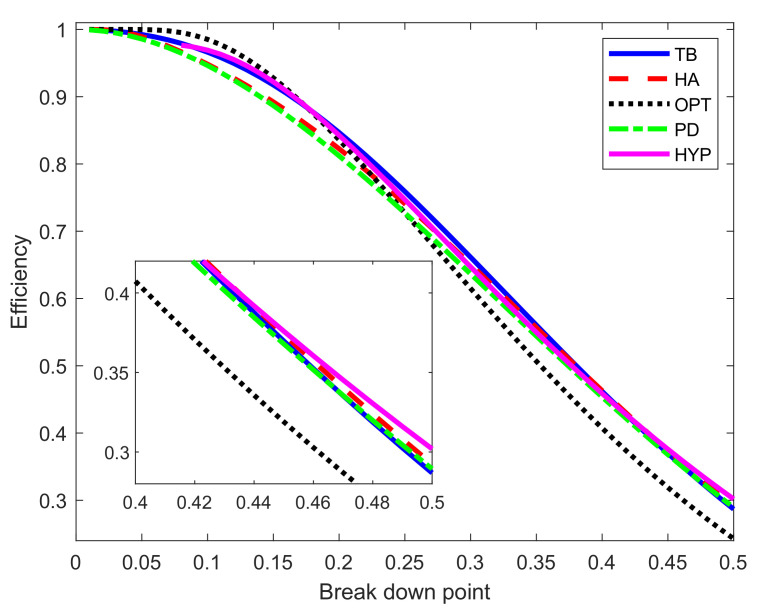
Breakdown point and efficiency as parameters vary for five rho functions: TB = Tukey biweight; HA = Hampel; OPT = optimal; PD = power divergence and HYP = hyperbolic. The inset is a zoom of the main figure for high breakdown point.

**Figure 6 entropy-22-00399-f006:**
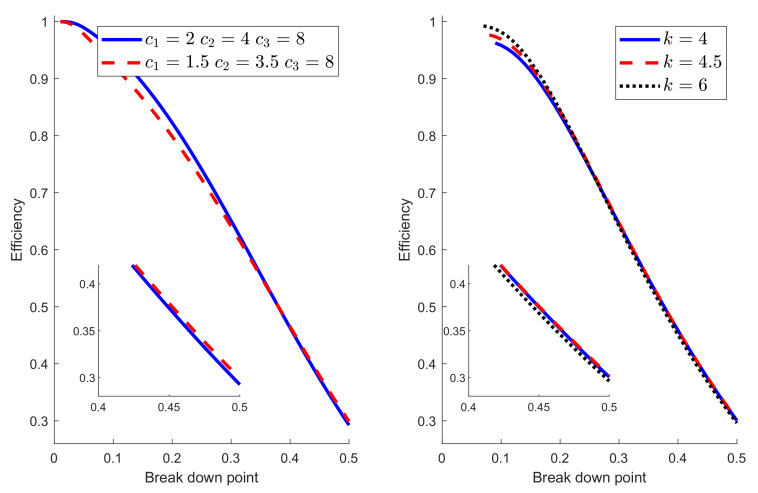
Breakdown point and efficiency as parameters vary for the Hampel and hyperbolic rho functions.

**Figure 7 entropy-22-00399-f007:**
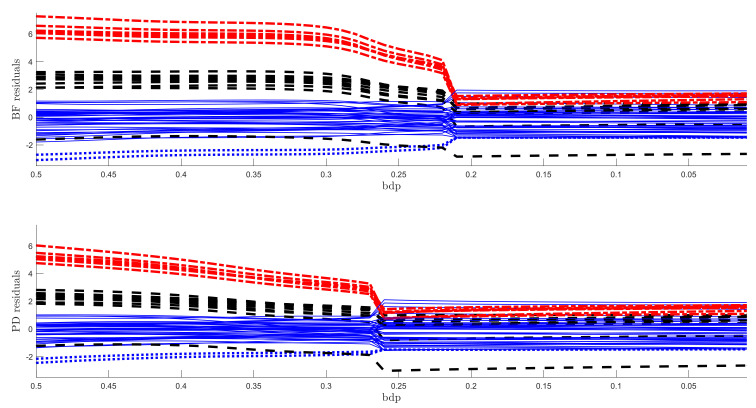
Regression data: residuals as bdp decreases. Upper panel, Brute Force (BF)-estimation, lower panel S-estimation.

**Figure 8 entropy-22-00399-f008:**
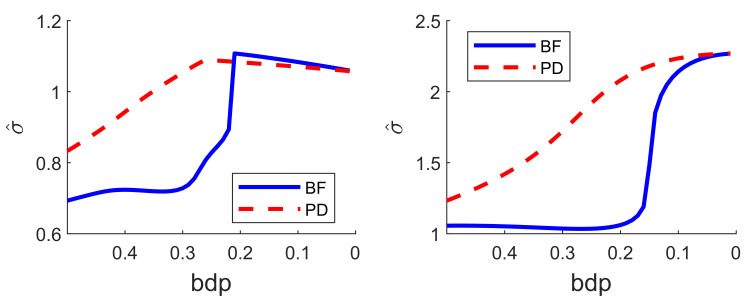
Comparison of estimates of σ as bdp decreases. Left-hand panel, regression data: right-hand panel, data with moderate outliers.

**Figure 9 entropy-22-00399-f009:**
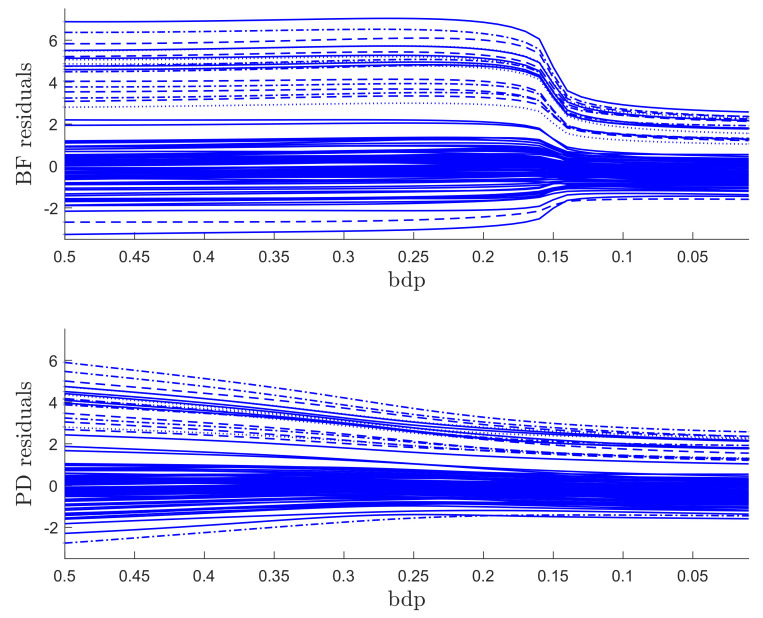
Data with moderate outliers: residuals as bdp decreases. Upper panel, BF-estimation; lower panel S-estimation.

**Figure 10 entropy-22-00399-f010:**
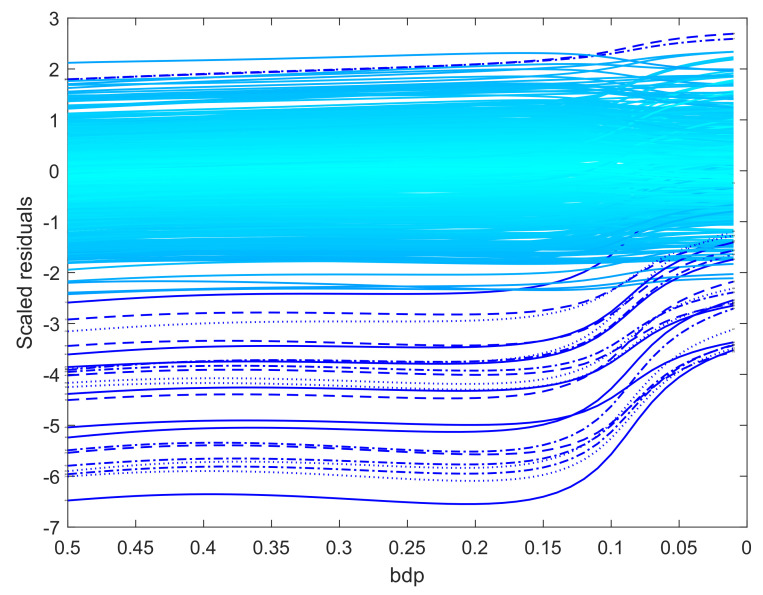
Loyalty card data: residuals for BF-estimation as bdp decreases.

**Table 1 entropy-22-00399-t001:** S power divergence. Values of α, bdp, and *eff* for three frequently used values of each in bold.

α	bdp	eff
**0**	0	1
**0.5**	0.1835	0.8381
**1**	0.2929	0.6495
0.7778	**0.25**	0.7271
1.7778	**0.4**	0.4536
3	**0.5**	0.2894
0.4715	0.1756	**0.85**
0.3522	0.14	**0.9**
0.2245	0.0963	**0.95**
0.089	0.0417	**0.99**

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
