# Peer review of "Robust Regression with Density Power Divergence: Theory, Comparisons, and Data Analysis"

_entropy, 2020, doi:10.3390/e22040399_

Round 1

Reviewer 2 Report

This is a very well presented article, building on a recent proposal by Basu, specific methodology and justifications of a general method in the important case of linear regression. This is a useful addition to the literature of Robust Statistics.

Author Response

Thank you very much for the careful reading of the paper and for your valuable comments which led to improve the current version of the paper

As concerns your point

In the appendix of the paper the authors consider  a number of rho functions for robust estimation and present their implementation in the FSDA toolkit. I wonder, if these functions are only suitable for regression or can also be applied for multivariate estimation of robust location and covariance?

In the appendix we have added the following sentence to clarify this issue

We have illustrated the use of the power divergence $\rho$ function in regression. But all these $\rho$ functions can also be used  for the estimation of robust location and covariance in the analysis of multivariate data. In this case, the scaled residuals $u$ are replaced by scaled Mahalanobis distances.

As concerns the minor points and corrections, you were absolutely right. We have corrected all of them

Reviewer 3 Report

The authors use the method of minimum density power divergency to define a rho function and apply it for S estimation in linear regression. The S-estimation theory allows to derive the asymptotic properties of the new estimates. These asymptotic properties are compared to a number of other rho and weight functions, but also the finite sample properties are compared, using the principle of monitoring robustness properties while changing tuning constants. 

The paper is very well structured, the presentation is clear and easy to follow. 

In the appendix of the paper the authors consider  a number of rho functions for robust estimation and present their implementation in the FSDA toolkit. I wonder, if these functions are only suitable for regression or can also be applied for multivariate estimation of robust location and covariance?

Minor comments and corrections:

  • P. 1, L. 25: write “... minimum density power divergence estimation ...”
  • P. 1, L. 26: write “... reviews S-estimation in the linear regression model and the second part......”
  • P. 2, L. 37: write BF instead of BS
  • P. 4, L. 1: write “The dispersion s is defined as …”

Author Response

(The authors gave the same response as above.)
